# Germination Behavior and Geographical Information System-Based Phenotyping of Root Hairs to Evaluate the Effects of Different Sources of Black Soldier Fly (*Hermetia illucens*) Larval Frass on Herbaceous Crops

**DOI:** 10.3390/plants13020230

**Published:** 2024-01-14

**Authors:** Rosanna Labella, Rocco Bochicchio, Rosangela Addesso, Donato Labella, Antonio Franco, Patrizia Falabella, Mariana Amato

**Affiliations:** 1School of Agriculture, Forestry, Food and Environmental Sciences, University of Basilicata, 85100 Potenza, Italy; rosannalabella@yahoo.it (R.L.); donato.labella@gmail.com (D.L.); 2Department of European and Mediterranean Cultures, Environment, and Cultural Heritage, University of Basilicata, 85100 Potenza, Italy; rosangela.addesso@unibas.it; 3Department of Science, University of Basilicata, 85100 Potenza, Italy; antonio.franco@unibas.it (A.F.); patrizia.falabella@unibas.it (P.F.); 4Spinoff XFlies s.r.l., University of Basilicata, 85100 Potenza, Italy

**Keywords:** phytotoxicity, chemical composition, salinity, nitrates, electrical conductivity

## Abstract

Insect larval frass has been proposed as a fertilizer and amendment, but methods for testing its effects on plants are poorly developed and need standardization. We obtained different types of black soldier fly (*Hermetia illucens)* frass via the factorial combination of (a) two insect diets, as follows: G (Gainesville = 50% wheat bran, 30% alfalfa meal, 20% maize meal) and W (43% sheep whey + 57% seeds); (b) two frass thermal treatments: NT = untreated and T = treated at 70 °C for 1 h. We tested the effects on the germination of cress (*Lepidium sativum* L.) and wheat (*Triticum durum* Desf.) by applying 1:2 w:w water extracts at 0, 25, 50, 75 and 100% concentration. Standardizing frass water content before extraction affected chemical composition. Frass extracts showed high electrical conductivity (8.88 to 13.78 mS cm^−1^). The W diet was suppressive towards *Escherichia coli* and showed a lower content of nitrates (e.g., WNT 40% lower than GNT) and a concentration-dependent phytotoxic effect on germinating plants. At 25% concentration, germination indices of G were 4.5 to 40-fold those at 100%. Root and shoot length and root hair area were affected by diet and concentration of frass extracts (e.g., root and shoot length in cress at 25% were, respectively, 4.53 and 2 times higher than at 100%), whereas the effects of the thermal treatment were few or inconclusive. On barley *(Hordeum vulgare* L.) grown in micropots on a silty loam soil, root mass was reduced by 37% at high extract concentration. A quick procedure for root hair surface area was developed based on the geographic information system (GIS) and may provide a fast method for incorporating root hair phenotyping in frass evaluation. The results indicate that below-ground structures need to be addressed in research on frass effects. For this, phyotoxicity tests should encompass different extract dilutions, and frass water content should be standardized before extraction in the direction of canonical procedures to allow comparisons.

## 1. Introduction

Agriculture can increasingly rely on a wealth of products issued from re-using, recycling and upcycling organic waste. Such products are quite diversified regarding nutritional and bio-stimulant properties for plants and may be suppressive toward selected pathogens. Among this new class of materials, insect larval frass is emerging [1,2]. Although the term frass would more properly refer to insect faeces, it is commonly used to designate the by-product of insect farming, including the remnants of larval exuviae, faeces, and undigested feed [3] after the separation from insect larvae, which may be destined to feed, energy, or green chemistry uses [4]. This by-product has been proposed as a fertilizer and amendment [5]. Research on frass effects on plants has just recently started, and one of the sources of frass that has raised interest issued from the farming of the bio-converter insect black soldier fly (*Hermetia illucens—Diptera: Stratiomyidae*) [2,3,4,5,6,7]. Chemical properties are affected by the rearing conditions, including insect diet [2], and show an interesting content of macro and micronutrients [1,2]; however, other compounds have been identified as potential bio-stimulants or exert antagonistic effects on pathogens. A typical compound found in frass is chitin [5] from larval exuviae, which is an amino-sugar polysaccharide that promotes plant growth and affects both beneficial and pathogen microbiota [8]. Potentially bioactive compounds have been identified in larvae and frass of black soldier fly [9,10]. 

First results on plant growth are encouraging, as reported by Guidini Lopes et al. [2] in a recent review. However, growth depression has also been documented, especially at high doses (e.g., [11,12,13]), and this has been attributed to scarce maturity in terms of knowledge concerning frass. In general, experiments have focused on above-ground plant parts, which are faster and easier to investigate. However, root growth is explicitly addressed in phytotoxicity germination tests. Specific phytotoxicity tests of frass have been performed with seed-germination-based trials on few frass types. Song et al. [13] show phytotoxicity in fresh frass 1:10 extract and a reduction of toxic effects after composting, whereas Setti et al. [12] did not find phytotoxicity in frass 1:20 extract. Liu et al. [14] found different levels of phytotoxicity in frass due to different larval diets and highlighted the role of ammonia, high electrical conductivity and volatile fatty acids in reducing germination indices. While very useful for a first approach to frass effects on plants, the results of these works are difficult to compare and generalize due to heterogeneity in methods. Limitations reside in differences or lack of detail regarding frass extract preparation procedures; also, germination tests have been performed with extract at one single concentration, with the further problem that it was not the same across works. Therefore, there is a need for standardization, which is common to other organic materials used in agriculture [15]. 

To the end of possible uses in agriculture, the current European regulations require that frass be subjected to thermal treatment at 70 °C for one hour (Reg Ue 2021/1925). One of the main expected outcomes of thermal treatments is to affect the microbiota. However, other processes like dehydration or volatilization will occur. This may affect plant behavior upon frass application. Therefore, recent research has been undertaken to investigate the effects of thermal treatments on frass effects on plant growth, and the initial evidence shows that tomato (*Solanum lycopersicon* L.) plants amended with frass from *H. illucens* subjected to thermal treatment exhibit lower weight than plants amended with untreated frass [16]. This first result needs to be explored further in the direction of assessing the effects of thermal treatments on the chemical composition of frass and above- and below-ground plant growth.

Root hairs are an important factor in plant health and behavior, and they represent a small percentage of the root mass (e.g., 2% [17]) but contribute as much as 67% to the total root surface [18]. Root hair density and length are linked to growth–media hospitability parameters such as water and nutrient content and microbial interactions [19] and, in turn, affect soil conditions [20] and crop behavior [21]. Root hair formation represents an important functional plant trait [22] and may, therefore, represent indicators of frass quality related to its effects on soil–plant relations. This feature may be especially relevant to describing the effects of microbial alterations due to frass thermal treatments. The quantification of root hairs poses methodological problems: it is mostly conducted through image acquisition of roots or their portions and subsequent manual or semi-automatic software-based analysis [23,24,25], but it is time-consuming since it often requires the manual delineation of root hair or root hair zone or training of algorithms or many segmentation steps to separate root hairs from other plant or background features [22,26]. Machine learning, deep learning, or neural networks have been proposed as root hair phenotyping methods but involve previous datasets and/or specialized software, or else models of root hair distribution [22,27]. 

The wide availability of software used to treat geographical information and the versatility of such systems [28] may prove useful for developing a tool for efficiently treating root hair images in order to quickly quantify the root hair parameters. This way, root hair phenotyping may become appealing in phytotoxicity/stimulation tests for frass agronomic evaluation. 

This work aims to test the effect of thermal treatment and different insect diets on frass properties and on effects on plant germination and emergence. We also tested the hypothesis that root hair formation and related features can provide an early indicator of frass quality. The novelty of this approach resides in adding a functional parameter relevant to plant–soil relations to phytotoxicity tests.

Two methodological aims of this work are as follows:
(i)to propose a protocol for standardizing germination tests for the evaluation of frass phytotoxicity, regarding both extract preparation and the use of multiple dilutions to highlight stimulation or toxicity on plant above- and below-ground growth;(ii)to develop a GIS (geographic information system)-based method for the quantification of root hair features. Applications of this method in root research would span from the fast evaluation of frass and other organic materials in germination tests to agronomic research on below-ground plant features.

## 2. Results

### 2.1. Chemical and Microbiological Properties of Frass and Frass Extracts

The chemical and microbiological compositions of the four frass types produced in our test are reported in Table 1.

The water content was lower (*p* < 0.05) in the thermally treated frass for both diets, and lower in G than W frass types. Organic matter and carbon were not affected by thermal treatment, but they were higher (*p* < 0.05) in W frass, which also showed a significantly higher percentage of N-NH_4_+ and a lower percentage of nitric forms of nitrogen. Nitrates were also significantly higher in thermally treated frass than in the untreated samples, but for the Gainesville diet only. Total nitrogen was higher in the treated frass for both diets, and the G samples scored higher values in terms of W for T only, whereas the reverse is true for NT. The thermally treated samples also showed a higher percentage of proteins and fibres. Carbohydrates were remarkably and significantly higher in the W diet, where fibre was undetectable. The G diet showed significantly higher values of ash, calcium, magnesium, potassium, and phosphorus, and lower amounts of lipids and sodium. Salmonellas were absent from all samples, whereas *Escherichia coli* was found in very low amounts in the W diet frass but at a remarkable concentration in G, where the thermal treatment reduced the number of bacterial colonies. Other bacteria were identified and mainly belong to Enterococcaceae in G, both thermally treated and untreated and *Aerococcus viridans* in W, regardless of thermal treatment. Electrical conductivity was significantly (*p* < 0.05) highest for WT followed by WNT, and then the G diet with both thermal treatments.

The chemical and microbiological composition of the AS (as such) and E (equilibrated) frass extracts are reported in Table 2 and Table 3, respectively.

The AS extracts (Table 2) did not show significant differences in terms of dry matter, organic matter and carbon between diet or thermal treatment. The highest total and ammoniacal nitrogen concentration was found in WT, whereas nitric nitrogen was significantly higher in the G diet. In general, the values of the nitrogen forms were higher in treated samples, except for nitric nitrogen in the W diet, where no significant difference was found between T and NT. The amount of fibre was quite low, and it was undetected in W samples. The results regarding elements and bacteria showed the same trends as in the solid frass samples (Table 1). Electrical conductivity values were higher than those of the solid frass and highest for WT. 

The extracts obtained using the E method (Table 3) were less concentrated than the AS extracts. They show lower dry matter and conductivity and lower concentration in terms of most variables. Differences and rankings between the treatments were different from those noted for AS extracts for several parameters; for instance, in E, the total nitrogen was higher for W than that seen in the G diet, without differences between the thermal treatments; the nitrates were not significantly different between the treated and untreated G, and lipids were higher in G, whereas the AS extracts showed higher nitrates in GT than in GNT and a higher lipid content in W.

### 2.2. Phytotoxicity Tests

The results from the phytotoxicity tests on frass from different sources extracted with the E method are reported in Figure 1, Figure 2, Figure 3, Figure 4, Figure 5, Figure 6, Figure 7, Figure 8 and Figure 9. Only statistically significant effects are shown. 

Figure 1 shows cress germination indices GERIS (shoot-based germination index) and GERIR (root-based germination index). Both cress germination indices were significantly affected by larval diet; they were higher than 100% for the G diet, whereas for the W diet, they showed values lower than 6%. 

The effect of the diet was shown by the significantly higher values of the number of germinated seeds (Figure 2a) and of shoot (Figure 2b) and root length (Figure 2d) for the G diet, except for root length at 100% concentration, where the values were not significantly different from those of the W diet. In addition, increasing the concentration of frass extracts significantly reduced shoot (Figure 2c) and root (Figure 2d) length despite the fact that differences between 25 and 50% concentrations did not reach statistical significance. The root length values were affected by concentration for the G diet only. At 75 and 100% concentration, no seeds germinated in the W diet.

The values of the number of germinated seeds and of root and shoot lengths (Figure 3) were significantly lower than those of the control in the W diet except for the untreated W at 25% for the number of germinated seeds. For the G diet, the values in terms of root length were significantly higher than those of the control for the treated frass extract at 25% and lower in a few instances at 100% concentration only.

For wheat, the G diet yielded values in terms of germination indices higher than those of the W diet, but a statistical significance in terms of differences was not reached for diet as a main effect (e.g., GERIR of G was 1.65 times that of W with *p* = 0.06). However, diet × thermal treatment interaction (Figure 4) shows that GERIS and the length of roots and shoots were significantly higher for G in thermally treated frass only.

The main effect of the concentration (Figure 5) showed statistically significant differences between 25%, which yielded values of GERIS and GERIR higher than 100%, and other concentrations that resulted in values of germination indices lower than 100%. The undiluted extract showed values lower than 7%.

The concentration effects were significant on the number of germinated seeds and the length of shoots and roots (Figure 6) where the 25% concentration was always significantly higher than the 100%, but differences between intermediate concentrations were not always significant.

The values of the number of germinated seeds and of root and shoot length (Figure 7) were significantly lower than those of the control at 100% concentration for all treatments and at lower concentrations in a few cases, especially for the W diet.

The emergence test conducted on barley showed a significant concentration effect on root dry weight (Figure 8a) and SPAD—measured Chlorophyll content (Figure 8b), where lower concentrations performed better than higher ones, though not all differences were significant.

The G diet proved significantly better than the W diet only for root/shoot mass at 25% concentration (Figure 8c), and the values decreased with extract concentration for both diets, although the differences were not always statistically significant.

Root mass (Figure 9a) was significantly lower than that of the control for all treatments at concentrations higher than 50%, except for GT, and the root:shoot ratio (Figure 9b) of the plants treated with frass extracts was lower than that of the control at all concentrations

### 2.3. Root Hairs 

Figure 10 and Figure 11 depict the root hair area projected on the plane of the images acquired on germinating seeds. The values are expressed through the sHa variable representing the surface area of root hairs per unit of root length. 

Examples of images corresponding to different root hair area values (Figure 10) show root hairs in the control treatment and in the W and G thermally treated samples at the same concentration of 25%. At concentrations of 50% and higher, no seeds germinated for W. Therefore, the factorial combination of diet and thermal treatment was tested using ANOVA on the 25% concentration only, and only the main effect of the diet was statistically significant, with higher values found upon the application of G extracts (Figure 11a). The factorial combination of concentration and thermal treatment was tested using ANOVA on the G diet only, and the concentration effects were significant (Figure 11b); root hair area per unit root length was higher in extracts at 25% concentration. The roots in the control treatment showed higher values of sHa than roots treated with frass extracts, but differences reached statistical significance for the GT treatment extracts at 100% concentration only and for the GNT at concentrations of 50% or higher. Differences between the control and samples from the W diet were significant at 25% concentration, and no comparison could be made for higher concentrations due to a lack of germinated seeds.

## 3. Discussion

### 3.1. Chemical Composition

In a review [2], a wide variation (even of orders of magnitude) in the chemical attributes of frass derived from black soldier fly larvae is reported as being related to the larval diet. Data from our experiment are within range but towards the low end for N values and the high end for C (in our case, organic carbon). The ratio of organic C/total N from our data ranges from around 26% in GT to about 40% in GNT. Setti et al. [12] report C/N values of eight for frass derived from *H. illucens,* suggesting that they are favorable for plant growth. Nevertheless, a wide range of C/N values (from 7.3 to 26.6) are reported by Guidini Lopes et al. [2], and this corresponds to very different situations as far as nitrogen availability is concerned. The C/N ratio of organic materials considered for use in agriculture as amendment and fertilizers usually ranges between 20 and 40 [2]. A rather high C/N ratio, as in our case, suggests that our specific frass types bear a higher potential value for amendment than as fertilizers, and the high end of the range of C/N found in frass corresponds to materials that may cause N immobilization when added to the soil. This effect is linked to microbial needs in terms of decomposing high-carbon soil organic matter. Therefore, frass solids with high C/N may result in stunted or slower plant growth and poorer nitrogen nutrition unless additional nitrogen fertilizer is added. 

The pH values of our frass are lower than average frass [2], especially so for our extracts. The electrical conductivity values found in our solid frass exceed values at which the yield of sensitive and moderately sensitive plants starts decreasing. Regarding the values we found in the AS and E extracts, they would cause yield reductions even in tolerant species [29,30]. The levels of many of the macro-elements and metals in our frass are lower than those reported for other black soldier fly frass produced when using the Gainesville diet [12]. 

We produced frass extracts using different methods. In our results, despite their equal dry matter content, significant differences between treatments were found in many parameters for both AS and E methods. Meanwhile, the amount, direction, and significance of differences were affected by the extraction method. The effects of extraction procedures on the chemical composition and electrical conductivity (EC) are reported in [31]. The E procedure allowed us to bring all samples to the same water content, therefore avoiding confounding effects due to superposing frass hydration differences with the composition of frass solids. This is useful for comparisons of different frass types and generalizations across literature reports. Therefore, we suggest adopting the E method in frass research for a clearer discussion of the results and comparison of the data.

Bacteria in frass may be harmful or beneficial for plants [32,33], but are sensitive to thermal treatment prescribed by legislations (EU Regulation 2021/1925) aimed at reducing risk due to pathogens like *Salmonella* spp., *E. coli* and others. In our case, all frass types did not contain *Salmonellas*, and this is in agreement with the literature [2].

We found significant and remarkable differences in *E. coli* levels in frass produced with different diets: whey–wheat residues fed to larvae showed a low (in extracts) or undetectable (in solids) presence of colonies compared to the G samples. In the latter diet, thermal treatment proved effective in reducing the pathogen’s presence. Other bacteria identified in our materials were different in frass produced with different larval diets. Variations in frass microbial composition with diet and growth conditions are also reported in the literature [34,35].

### 3.2. Phytotoxicity: Germination and Emergence Tests

In our data, the phytotoxicity results showed the effects of diet and the concentration of frass extracts, whereas the effects of thermal treatment were not significant in many cases or did not show clear trends. As a main effect, the G diet showed stimulation, whereas the W diet proved phytotoxic, especially for cress (Figure 1). One or more parameters of germination or seedling growth were favored in terms of the application of diluted extracts at 25% concentration, whereas, at higher concentrations, they showed values lower than the controls and decreased in a dose-dependent way. Our experimental factors showed the same general trends in emergence tests but with less striking—and not always significant—effects. The below-ground parameters proved more sensitive than shoot mass to frass extract application in the soil. The effects of frass concentration on chlorophyll content were also found.

Setti et al. [12] take a 50% reduction in germination indices as the phytotoxicity threshold following the pioneering work of Zucconi et al. [36], whereas other sources indicate a threshold of 70% [31] or 80% [37]. In our data, the values of the germination indices of the W diet were lower than both of these thresholds for cress and for concentrations higher than 50% for wheat. The literature reports on frass phytotoxicity are few and contradictory; germination index values much lower than 50% have been found in fresh frass 1:10 extract applied on pak choi (*Brassica rapa*) [13], but phytotoxicity was reduced after aerated composting. Setti et al. [12] did not find phytotoxicity in a frass 1:20 extract since the germination index values were higher than 100% and greater for roots than for shoots of cress. Liu et al. [14] show that composting with black soldier fly larvae improves germination indices compared to the starting materials and found that two out of three frass types were not phytotoxic based on a 1:10 extract ratio calculated on frass dry weight. Liu et al. [37] found phytotoxicity in one out of two cases. A lack of standardization in terms of phytotoxicity tests based on seed germination is a common problem for organic matrices like compost [15]. As discussed in the chemical composition section, we used two extraction procedures: AS, where different types of frass were used at their initial water content, and the E procedure, where the extraction was performed after bringing all frass types to the same water content in order to discount the concentration effects due to the initial frass hydration conditions. It has been reported that an underestimation of phytotoxicity may occur if frass moisture is high or an overestimation if frass has a low water content [31], and this is a further indication supporting the need for standardization. In our procedure, we used a low extraction ratio (1:2) after bringing all frass types to an 85% water content and used dilutions from 100 to 25% to provide a range of conditions where both stimulation and inhibition may occur.

The phytotoxicity of frass extracts has been discussed in terms of lack of maturity [11,13,14], but identifying specific elements is not straightforward. High electrical conductivity and volatile fatty acids have been listed as important factors for reduced germination indices upon the application of frass [14], but interactions with other frass properties is also relevant. Phytotoxicity tests are based on the global responses of plants, and therefore, may be due to several factors. In our case, low-dose stimulation may be ascribed to nitrogen and phosphorus as well as microbial growth promotion, especially for the G diet samples. Otherwise, the high-dose depression of growth found in our extracts—stronger for cress than wheat—may be linked to the high electrical conductivity of our extracts. In our data, this was to found to be 4–5 mS cm^−1^ for frass solids 12–15 mS cm^−1^ for AS, and 8.88 to 9.76 mS cm^−1^ for the E method, which are high enough to inhibit plant performance [29,30]. Especially poor performances are found with W frass extracts for *L. sativum,* which is classified as a glycophyte with low tolerance to salt stress [38], whereas better growth was found for wheat, which is classically ascribed to the moderate tolerance of the species [29,30].

Higher phytotoxicity was found in W than in G, and this can be explained by a nitrate–salinity interaction rather than simply due to the differences in electrical conductivity that we recorded. Salinity is a well-known factor of reduced germination [39], while nitrate has been reported to stimulate germination in a wide variety of plant species [40] and improve germination (seed-priming effect) in adverse conditions [41], specifically salinity [42]. In our data, the salinity and nitrate contents of the extracts are generally different between the G and W diets. Electrical conductivity is quite high, but in solids and AS extracts, nitrates may be high enough—and sodium low enough—to overcome salinity, especially for the G diet. The E extract showed a much lower concentration of nitrates and macronutrients than solids and AS while also still exhibiting high conductivity; for instance, E nitrates are 43 to 133% lower than in AS, while electrical conductivity is 27 to 48% lower, and it still exhibits values of around 9 mS cm^−1^, which are harmful to most plant species. We may hypothesize that in E extracts, salinity was high enough to overcome the effect of nitrates and macronutrients, whereas, if diluted (e.g., 25% or 50% dilutions), harmful salinity effects decrease, and even a low concentration of nutrients is enough to stimulate the germination and elongation of roots and shoots compared to the control. Other toxic elements in frass and its extracts, such as sodium, especially in the W diet, may have similar effects in terms of salinity. The G treatment also shows higher percent values of other macronutrients and lower Na than W, and all this contributes to better plant performances.

Regarding the thermal treatment, we have seldom found statistical significance in terms of the effects and results, providing no clear indication. 

Other than salinity or nutrient content, the effects of frass on plant growth have been ascribed to bio-stimulant components [2], which are not the object of this study. However, we showed a higher presence of *E. coli* in G frass, especially so in non-thermally treated frass samples. *E. coli* is not only a potentially dangerous pathogen but has also been shown to mutualistically colonize plant roots and improve root growth and photosynthetic pigments, which is likely due to indolacetic acid [43]. Our results that suggest better plant performance in G may, therefore, be ascribed, at least in part, to the role of this and other plant growth-promoting bacteria in the stimulation of plant growth. Nevertheless, in our results, the effects of thermal treatment on the germination indices and plant emergence were not conclusive; therefore, differences in the presence of growth-promoting bacteria do not seem relevant in our case. 

The dose dependance of our results indicates that different doses and dilutions need to be used in toxicity tests for frass.

### 3.3. Root Hairs

Our data show a reduction in root hair area per unit of root length in wheat seeds treated with frass extracts compared to the control in a diet and dose-dependent way, although the differences were not always significant. Based on our results, we can identify differences in root hair area between the G and W diets at the 25% extract concentration and between concentrations for the G diet. Root hairs are an important trait, which may be more relevant for plant biomass accumulation than mycorrhizal association in some conditions [19]. They have been identified as key features for nitrate acquisition and plant growth [44].

Therefore, our data show that frass application during seed germination do not affect root length alone but rely on traits that impact root functionality. Our results suggest that the germination tests used to assess the phytotoxicity of frass might be improved via additional root hair determination. This would widen the picture from the measurement of plant growth to the determination of phenes that may, in turn, play a role in furthering the ability of plants to acquire resources and deal with environmental constraints.

Methods for measuring root hair features are based on the separation of root hairs and other root or background features via manual or semi-automatic segmentation or machine learning techniques. In some cases, lengthy procedures and criteria are justified by the need to discriminate root hairs from a heterogeneous background, such as the surrounding soil in “in vivo” determinations [26]. Additionally, the most appropriate root hair features need to be established for specific research objectives [22]. In our case, the fast reclassification of images, the extraction of root hairs and the measurement of surfaces conducted with GIS software is a promising technique for the purposes of complementing germination tests for assessing the phytotoxicity/stimulation of frass or other organic materials, where observations are conducted on roots growing in simplified media, such as germination paper. 

## 4. Materials and Methods

Three plant species were used: wheat (*Triticum durum* Desf.) cv. Ciclope, barley (*Hordeum vulgare* L.) cv. Laureate, and cress (*Lepidium sativum* L.). 

Four different larval frass types identified as GT, GNT, WT, and WNT were obtained after breeding black soldier flies (*Hermetia illucens*) with a factorial combination of methods, namely the following: 

(a) insect diet with two levels: G = Gainesville diet (50% wheat bran, 30% alfalfa meal, and 20% maize kernels) and W = whey–seed diet (43% sheep whey + 57% wheat and segetal species seeds); 

(b) thermal treatments with two levels: untreated (NT) and treated at 70 °C for 1 h (T).

### 4.1. Frass Properties and Extract Preparation

At the end of the breeding + thermal treatment processes, the water content was gravimetrically measured by weighing samples before and after oven-drying at 70 °C, and different average percent values were found for the different frass types (see Table 1). 

The Frass extracts were prepared following two methods:

AS = as such. Frass was mixed with water at a 1:5 w:w frass:water ratio, shaken for 40 min and centrifugated at 6000 rpm for 15 min at 20 °C.

E = equilibrated. All frass types were brought to 85% water content and extracted with a 1:2 w:w frass/water ratio after shaking for 40 min and centrifugated at 6000 rpm for 15 min at 20 °C.

The supernatant was collected and used for chemical and microbiological analyses and used for the experiments.

The following chemical and microbiological analyses were outsourced and performed in triplicate at the ARA (Associazione Regionale Allevatori) laboratory using the following methods:

For the dry matter of the extracts, a gravimetric examination was conducted upon oven drying at 105 °C. For the water content and dry matter of solid frass, protein, fibre, carbohydrates and lipid concentrations, near-infrared (NIR) spectrometry with a NIRS DS3 analyzer was used (FOSS—Hilleroed, Denmark). For the total nitrogen, the Kjeldhal method was used [45]. For N-NH_4_^+^ and N-NO_3_-, the Ulsch method was used [46]. Organic carbon was examined using the Springer–Klee method [47]. Organic matter was calculated based on organic carbon, as follows: Organic matter (%) = Total organic carbon (%) × 1.724. pH and electrical conductivity were measured in the extracts as such and in solids in a 3:50 w:w frass:water ratio slurry. Ash was examined using a muffle furnace (Model F1A00010 (ISCO s.r.l., Milan, Italy) at 550 °C. Elements Ca, K, Mg, P, Na, Cd, Ni, Pb, Zn, and As were studied using Inductively Coupled Plasma–Optical Emission Spectroscopy (ICP-OES 5100—Agilent Technologies, Santa Clara, CA, USA). Microbiological determinations were conducted using International PBI (Milan, Italy) incubators with culture media from Biolife Italiana (Milan, Italy) and bioMérieux Italia (Baagno a Ripoli, Italy) following the ISO 6579-1:2017 international standard method for *Salmonella* [48] and the ISO 16649-2:2001 international standard method for *E. coli* [49].

### 4.2. Phytotoxicity Test

Phytotoxicity tests were performed using different dilutions of frass extract. 

An extract test achieved with the E method described in Section 4.1 was performed on wheat and cress with frass extracted from treatments undertook with the following experimental design:

(a) frass from insect diet with two levels: G and W; 

(b) frass from thermal treatments with two levels: NT and T;

(c) a 1:2 *w*/*w* frass extract applied at 4 dilution levels: 25%, 50%, 75% and 100%.

The control treatment consisted of distilled water.

Wheat seeds were selected for weight uniformity (between 55.00 and 57.00 mg). They were surface-sterilized with 5% Na-hypochlorite solution for 2 min, and then rinsed three times with sterile H_2_O. Cress seeds were not surface-sterilized in order not to let mucilage be extruded before the beginning of the experiment. 

Three replicated Petri dishes per treatment were lined with 90 mm diameter Whatman n.1 paper and filled with 10 kernels (for wheat and barley) or 20 cress seeds and soaked with 5 mL of distilled water or a water extract of frass obtained with the methods and experimental designs described above. 

After germination in the dark at the ambient temperature for 72 h for cress and wheat and 120 h for barley, germination indices were determined; the length of the roots and shoots were measured on 30 seeds in cereals and 60 seeds for cress for each treatment.

Phytotoxicity was evaluated through the following indexes:

Germination index for roots (GERIR):GERIR% = 100 × G1/G2 × R1/R2

Germination index for shoots (GERIS):GERIS% = 100 × G1/G2 × S1/S2
where G1 and G2 are germinated seeds in the sample and control; R1 and R2 are mean root length for the sample and for the control, respectively; and S1 and S2 are mean root length for the sample and for the control, respectively. 

### 4.3. Emergence Tests

The barley seedlings were grown in micropots (10 mL volume) filled with a silty loam soil with the following characteristics: sand (50–2000 μm) 43.6%, silt (2–50 μm) 34.2%, clay < 2 μm) 22.1%., and pH 6.8. The pots were filled with 9 g of soil irrigated to the field capacity of 27.9%, which was determined as the gravimetric percent of water retained at −0.33 bars of suction on a porous plate by adding control water or the 1:2 E frass extract at the concentrations described in Section 4.1. 

The seeds were pre-germinated for 48 h and planted in 9 replicated micropots, where they grew at an average temperature of 20 °C until the first leaf was produced.

At the end of the experiment, the following measurements were performed:
-The chlorophyll content was measured on three leaves per plant with a leaf transmittance leaf clip chlorophyll concentration meter (MC-100 Apogee instruments— Inc., Logan, UT, USA) and converted into SPAD (Soil Plant Analysis Development) units [50] as the difference between leaf transmittance at the red light wavelength of 653 nm and leaf transmittance at the near-infrared wavelength of 931 nm. -Biometric measurements were performed as follows: the above-ground dry mass was determined after oven drying at 70 °C until reaching a constant weight. The soil–root complex was gently extracted from the micro-pots by washing over a 0.5 mm mesh and oven-dried at 70 °C until reaching a constant weight to obtain the root dry mass. 


We then calculated the root-to-shoot biomass ratio (g g^−1^).

### 4.4. Root Hairs 

The treatments and growth conditions described for the dilution test in Section 4.1 were applied to wheat seeds for the determination of root hairs on four replicated seeds after 65 h of incubation.

Roots close to the germinating seeds were photographed at 8 × magnification with a Leica EZ4W microscope. Depending on root straightness, this corresponded to 9.9 to 12.7 mm root segments, and a 2 mm scale segment was included in each image. Image analysis was performed using ArcGIS version 10.8 (ESRI, Redlands, CA, USA) software:

RGB images were converted to floating point representation, and the images were reclassified in 10 classes according to the Natural Breaks Jenks clustering model [51]. For the resulting 10-color images, pixels with color(s) corresponding to root hairs were extracted with a conditioned query. The format was then converted from raster (pixel) to vector (polygons), and the area of the surface occupied by extracted polygons was determined and named Ha = root hairs area were projected on the image plane (mm^2^). During the latter step, images were manually cleaned by deleting false positives identified as areas extracted with root hairs but not attached to root structures. Images from the described steps of image analysis are reported in Figure 12. The specific surface area of root hairs (sHa) (mm^2^ mm^−1^) was calculated by dividing Ha by the length of the root in the image.

### 4.5. Statistical Analysis

Statistical analysis in the univariate domain was performed using the R programming language [52], with functions from ‘stats’ and ‘agricolae’ packages. The differences between the treatments from the factorial combination of (a) insect diet, (b) thermal treatment, (c) dilution levels and their interactions were analyzed with 3-way ANOVA, and mean separation was performed using Tukey’s post hoc test at *p* = 0.05. Where a control treatment of distilled water was present, differences between the control and each of the 16 treatments issued from the factorial combination of factors (a), (b) and (c) were tested using an unpaired *t*-test.

## 5. Conclusions

By varying the diet of *Hermetia illucens* larvae, we produced frass types with different effects on plants; in general, the above- and below-ground growth of seedlings in germination and emergence experiments was reduced if extracts of frass from a whey + seeds-based diet (W) were applied, whereas frass from the Gainesville (G) diet had a stimulating effect on some plant features at low extract concentrations but it showed phytotoxicity at high concentrations. The dilution-dependent behavior of plants treated with frass extracts was observed for most of the measured plant traits. Standardizing frass water content before extraction affected the chemical composition of the extracts. Based on our data, frass water content should be standardized before extraction, and phytotoxicity tests and plant trials should encompass different extract dilutions in the direction of canonical procedures to allow comparisons.

Emergence tests showed a lower sensitivity of plants to frass extract treatment than germination tests, and below-ground parts proved more responsive. We developed a fast procedure for root hair area determination based on GIS software. Root hair surface area per unit root length proved sensitive to frass extract application. The relevance of this finding resides in the fact that not only the length of the roots but also their functionality may be phenotyped in phytotoxicity tests, and adding this trait to simple growth parameters would increase the power of germination tests in general. This way, the interpretation of the effects of any chemical or physical treatment on germinating seeds would go beyond recording a slower or smaller growth by including a feature that indicates the plant’s ability to interact with the soil environment in its future stages of development.

Overall, our results indicate that roots and related attributes need to be addressed in research on frass effects, and our quick method may overcome some of the existing methodological limitations for root hair determination thanks to speed and the use of widespread non-specialized software.

The effects of frass on plants corresponded to different levels of chemical properties of our frass types. We identified possible determinants of plant behavior in the frass levels of electrical conductivity, nitrates and phosphorus and their interactions. Nevertheless, our interpretation is limited by the lack of more specific studies on bio-stimulating or toxic compounds; therefore, we cannot conclude that the causes of plant growth promotion and phytotoxicity were fully characterized in our study, and the mechanisms were not investigated. Therefore, further studies are needed for a more thorough characterization and search for molecules active in plant processes. To this end, we may envisage the use of advanced chemical techniques such as X-ray diffraction, Fourier-transform infrared or ultraviolet-visible spectroscopy, gas/liquid chromatography–mass spectrometry, and nuclear magnetic resonance. 

Additionally, future directions of research include plant–microbe interactions, which may better clarify the relevance of frass thermal treatment on plant growth. In our study, we highlighted some differences (e.g., in nitrate concentration) linked to thermal treatment, but plant responses were inconclusive. 

Our frass and extracts showed rather low N concentration in relation to C values and, therefore, may be of lower relevance as fertilizers than for soil amendment. Future research may investigate this by addressing soil physical properties as a consequence of frass or frass extract application.

## Figures and Tables

**Figure 1 plants-13-00230-f001:**
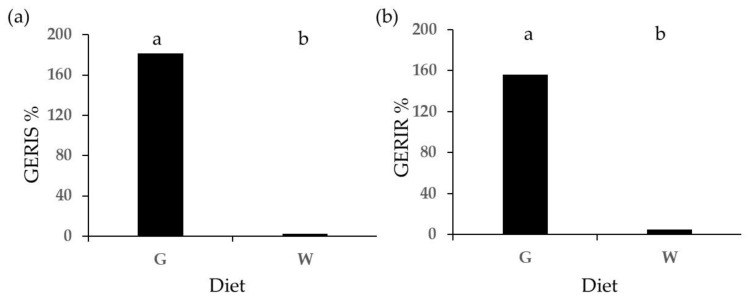
Diet effect on germination indices of cress treated with frass extracted with the E method: (**a**) GERIS; (**b**) GERIR. Different letters designate significantly different values for *p* < 0.05 at the Tukey’s post hoc mean separation test.

**Figure 2 plants-13-00230-f002:**
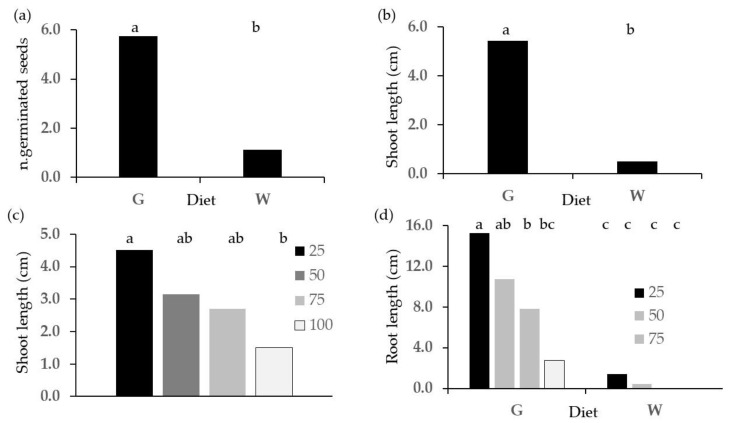
Effects of diet and concentration of frass extracted with the E method on germinating cress seeds: (**a**) diet effect on n. of germinated seeds; (**b**) diet effect on length of shoot; (**c**) extract concentration effect on length of shoot; (**d**) diet concentration interaction on root length. Different letters designate significantly different values for *p* < 0.05 at the Tukey’s post hoc mean separation test.

**Figure 3 plants-13-00230-f003:**
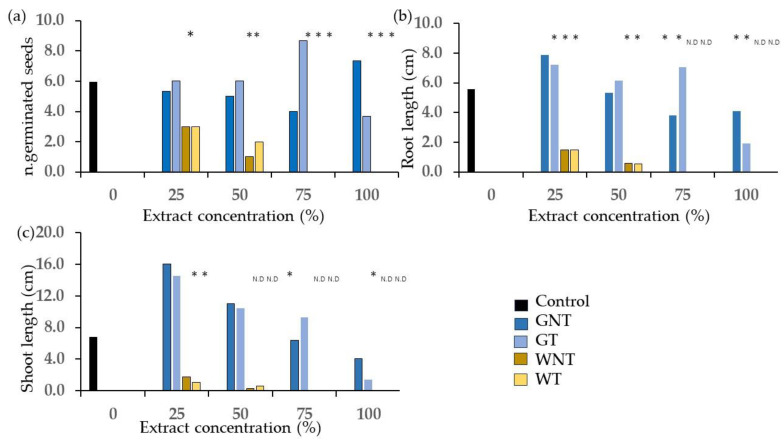
Comparison of control and treatments from the factorial combination of diet, thermal treatment and concentration of frass extract on germinating cress seeds: (**a**) n. of germinated seeds; (**b**) root length; (**c**) shoot length. N.D. = values not determined due to lack of germinated seeds. Asterisks designate values that are significantly different at *p* < 0.05 from the control at the unpaired *t*-test.

**Figure 4 plants-13-00230-f004:**
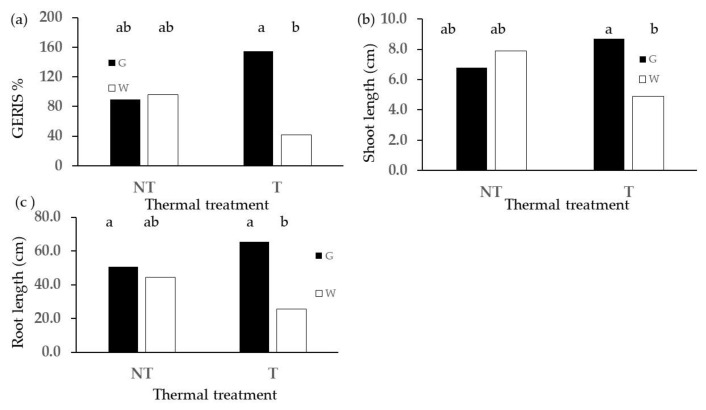
Effect of diet × thermal treatment interaction on (**a**) GERIS, (**b**) length of shoot, (**c**) root length. Different letters designate significantly different values for *p* < 0.05 at the Tukey’s post hoc mean separation test.

**Figure 5 plants-13-00230-f005:**
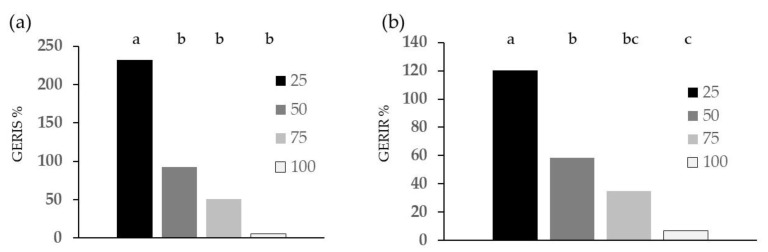
Effect of concentration of frass extracts on wheat germination indices: (**a**) GERIS; (**b**) GERIR. Different letters designate significantly different values for *p* < 0.05 at the Tukey’s post hoc mean separation test.

**Figure 6 plants-13-00230-f006:**
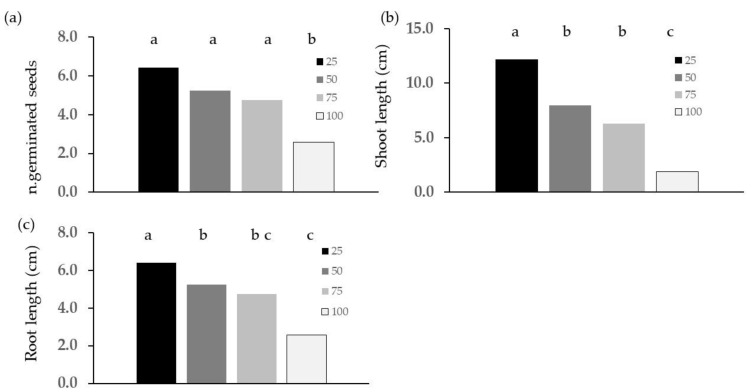
Effect of concentration of frass extracts on wheat germinating seeds: (**a**) n. of germinated seeds; (**b**) length of shoot; (**c**) root length. Different letters designate significantly different values for *p* < 0.05 at the Tukey’s post hoc mean separation test.

**Figure 7 plants-13-00230-f007:**
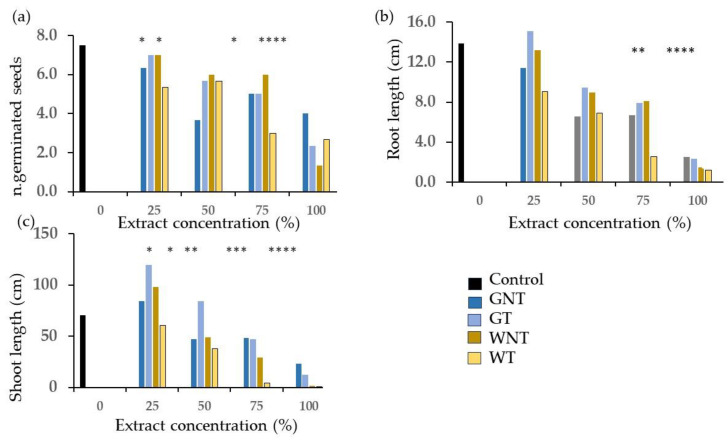
Comparison of control and treatments from the factorial combination of diet, thermal treatment and concentration of frass extract on wheat germinating seeds: (**a**) n. of germinated seeds; (**b**) root length; (**c**) shoot length. Asterisks designate values that are significantly different at *p* < 0.05 from the control at the unpaired *t*-test.

**Figure 8 plants-13-00230-f008:**
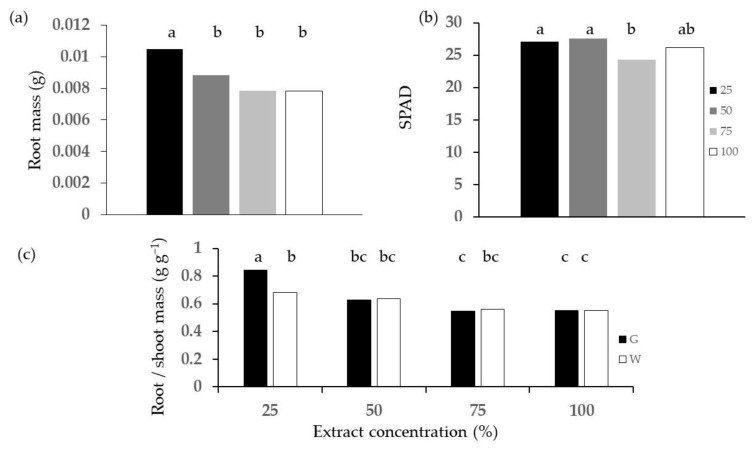
Effect of concentration of frass extracts on barley emergence: (**a**) root dry weight (**b**) SPAD; (**c**) root:shoot mass ratio. Different letters designate significantly different values for *p* < 0.05 at the Tukey’s post hoc mean separation test.

**Figure 9 plants-13-00230-f009:**
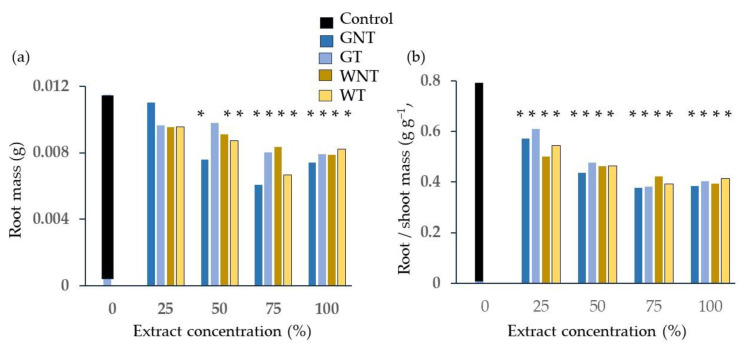
Comparison of the control and treatments from the factorial combination of diet, thermal treatment and concentration of frass extract on barley emergence: (**a**) root mass; (**b**) root:shoot ratio. Asterisks designate values that are significantly different at *p* < 0.05 from the control in terms of the unpaired *t*-test.

**Figure 10 plants-13-00230-f010:**
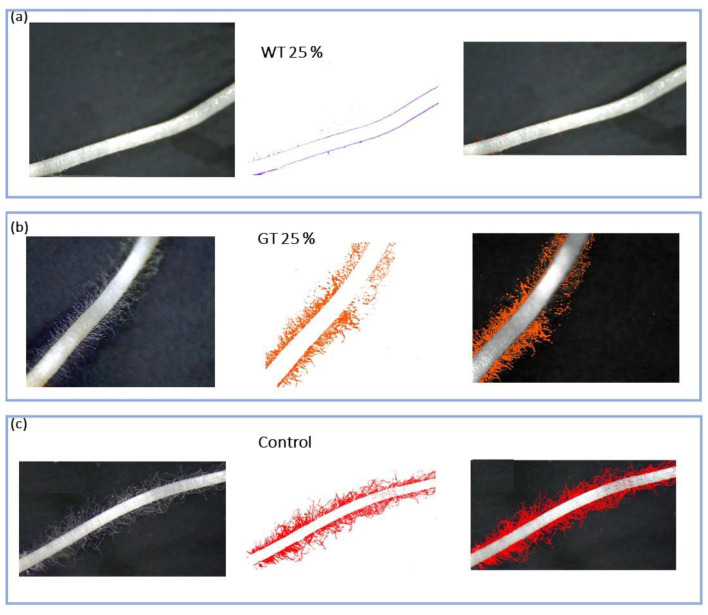
Examples of root hair images for (**a**) thermally treated W diet frass extract at 25% concentration; (**b**) thermally treated) G diet frass extract at 25% concentration; (**c**) control. **Left**: RGB images at 8 × magnification. **Center**: images where root hairs have been extracted after reclassification according to Jenk’s model. **Right**: images after vectorization, as described in the Materials and Methods section.

**Figure 11 plants-13-00230-f011:**
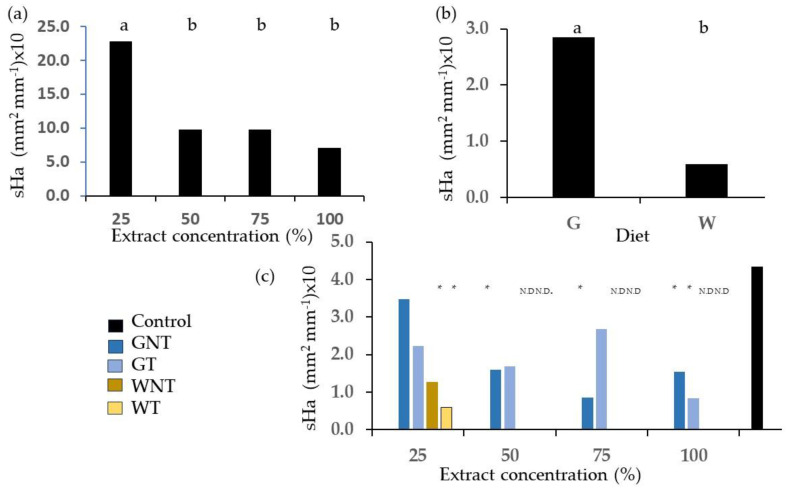
Surface area of root hairs projected on the image plane per unit root length: (**a**) diet effect at the 25% concentration of extracts; (**b**) concentration effect in G diet extracts. Different letters designate significantly different values for *p* < 0.05 at the Tukey’s post hoc mean separation test; (**c**) comparison of the control and treatments from the factorial combination of diet, thermal treatment and concentration of frass extract on wheat root hairs. Asterisks designate values that are significantly different at *p* < 0.05 from the control upon unpaired *t*-test.

**Figure 12 plants-13-00230-f012:**
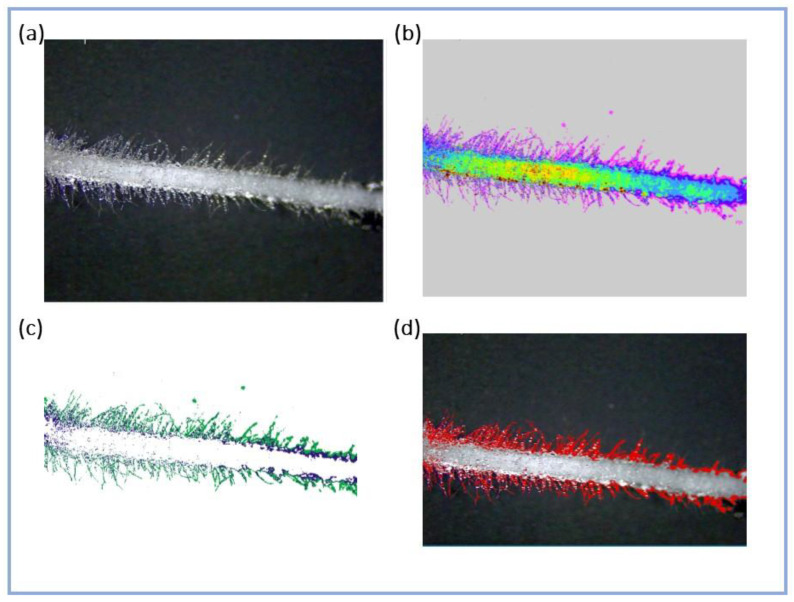
The steps of root hairs area measurement are as follows: (**a**) RGB image photographed at 8× magnification; (**b**) reclassified image according to Jenks’ model; (**c**) extraction of values corresponding to root hairs; (**d**) vectorization and manual cleaning.

**Table 1 plants-13-00230-t001:** Chemical and microbiological compositions of larval frass types issued from different larval diets. Concentration data refer to dry matter. Results are the average of three repeats ± standard deviation. Different letters designate significantly different values for *p* < 0.05 at Tukey’s post hoc mean separation test. GNT = frass from the Gainesville diet not treated thermally; GT = thermally treated frass from the Gainesville diet; WNT = frass from the whey + seeds diet not treated thermally; WT = thermally treated frass from the whey + seeds diet.

	Water Content	Dry Matter	Total N	Organic Matter	Organic Carbon
	**%**	**%**	**%**	**%**	**%**
GNT	19.53 ∓ 0.058 ^b^	80.47 *±* 0.058 ^c^	1.97 *±* 0.015 ^a^	88.10 *±* 0.548 ^b^	51.10 *±* 0.318 ^b^
GT	36.00 *±* 0.000 ^a^	64.00 *±* 0.000 ^d^	1.28 *±* 0.050 ^d^	88.83 *±* 0.277 ^b^	51.52 *±* 0.161 ^b^
WNT	13.40 *±* 0.000 ^d^	86.60 *±* 0.000 ^a^	1.86 *±* 0.015 ^b^	93.53 *±* 0.200 ^a^	54.25 *±* 0.116 ^a^
WT	15.10 *±* 0.000 ^c^	84.90 *±* 0.000 ^b^	1.60 *±* 0.017 ^c^	93.72 *±* 0.142 ^a^	54.36 *±* 0.082 ^a^
	N-NH_4_^+^	N-NO_3_-	pH	Conductivity	Protein
	mg/Kg	mg/Kg		mS cm^−1^	%
GNT	4741.0 *±* 3.61 ^c^	3.82 *±* 0.010 ^a^	7.20 *±* 0.000 ^c^	4.63 *±* 0.039 ^b^	11.15 *±* 0.241 ^a^
GT	4770.1 *±* 0.51 ^c^	3.02 *±* 0.007 ^b^	7.04 *±* 0.000 ^d^	4.56 *±* 0.040 ^b^	7.73 *±* 0.040 ^c^
WNT	5980.3 *±* 1.53 ^b^	2.47 *±* 0.013 ^c^	7.32 *±* 0.000 ^b^	5.09 *±* 0.000 ^a^	11.60 *±* 0.273 ^a^
WT	6270.0 *±* 4.00 ^a^	2.49 *±* 0.020 ^c^	7.64 *±* 0.000 ^a^	4.60 *±* 0.035 ^b^	10.13 *±* 0.260 ^b^
	Fibre	Carbohydrates	Lipids	Ash	Ca
	%	%	%	%	mg/Kg
GNT	14.25 *±* 0.006 ^a^	2.27 *±* 0.261 ^b^	1.46 *±* 0.021 ^b^	9.58 *±* 0.108 ^a^	4412.0 *±* 1.00 ^a^
GT	11.94 *±* 0.025 ^b^	2.68 *±* 0.345 ^b^	1.16 *±* 0.015 ^c^	7.15 *±* 0.126 ^b^	2064.7 *±* 4.51 ^b^
WNT	<0.01	23.43 *±* 0.310 ^a^	1.61 *±* 0.024 ^a^	5.60 *±* 0.160 ^c^	1317.4 *±* 0.47 ^c^
WT	<0.01	22.82 *±* 0.360 ^a^	1.56 *±* 0.030 ^a^	5.33 *±* 0.183 ^c^	1044.7 *±* 0.46 ^d^
	K	Mg	Na	P	Cd
	mg/Kg	mg/Kg	mg/Kg	mg/Kg	mg/Kg
GNT	19,276.0 *±* 201.85 ^a^	7152.2 *±* 162.45 ^a^	974.1 *±* 39.17 ^b^	9592.5 *±* 31.07 ^a^	0.136 *±* 0.115 ^b^
GT	19,378.0 *±* 187.72 ^a^	6833.3 *±* 152.81 ^a^	987.8 *±* 46.08 ^b^	9670.3 *±* 40.25 ^a^	0.120 *±* 0.053 ^b^
WNT	11,698.1 *±* 193.17 ^b^	3765.5 *±* 136.17 ^b^	2022.3 *±* 42.39 ^a^	5288.0 *±* 39.00 ^b^	0.124 *±* 0.056 ^b^
WT	11,333.0 *±* 168.49 ^b^	3679.8 *±* 112.50 ^b^	1944.4 *±* 49.17 ^a^	5039.4 *±* 40.41 ^b^	0.535 *±* 0.030 ^a^
	Ni	Pb	Zn	As	*Escherichia coli*
	mg/Kg	mg/Kg	mg/Kg	mg/Kg	UFC/g
GNT	49.533 *±* 4.11 ^a^	0.901 *±* 0.010 ^a^	93.167 *±* 2.20 ^a^	0.228 *±* 0.037 ^a^	>150,000
GT	42.033 *±* 2.58 ^a^	0.714 *±* 0.014 ^b^	89.267 *±* 2.11 ^b^	0.243 *±* 0.021 ^a^	2567 *±* 208.17
WNT	27.200 *±* 2.06 ^b^	1.671 *±* 0.003 ^c^	55.167 *±* 3.05 ^c^	0.133 *±* 0.022 ^b^	<10
WT	25.200 *±* 1.01 ^b^	0.819 *±* 0.003 ^d^	50.600 *±* 2.91 ^c^	0.128 *±* 0.014 ^b^	<10

**Table 2 plants-13-00230-t002:** Chemical and microbiological composition of the water extracts of larval frass types issued from different larval diets and extracted using the AS method. Concentration data refer to dry matter. The results are the average of three repeats ± standard deviation. Different letters designate significantly different values for *p* < 0.05 at the Tukey’s post hoc mean separation test. GNT = frass from Gainesville diet not treated thermally; GT = thermally treated frass from Gainesville diet; WNT = frass from the whey + seeds diet not treated thermally; WT = thermally treated frass from the whey + seeds diet.

	Water Content	Dry Matter	Total N	Organic Matter	Organic Carbon
	%	%	%	%	%
GNT	97.00 *±* 0.351 ^a^	3.00 *±* 0.351 ^a^	0.0061 *±* 0.0003 ^b^	98.98 *±* 0.402 ^a^	57.41 *±* 0.233 ^a^
GT	97.10 *±* 0.252 ^a^	2.90 *±* 0.252 ^a^	0.0039 *±* 0.0003 ^c^	99.07 *±* 0.499 ^a^	57.46 *±* 0.290 ^a^
WNT	97.50 *±* 0.153 ^a^	2.50 *±* 0.152 ^a^	0.0088 *±* 0.0002 ^a^	99.34 *±* 0.204 ^a^	57.62 *±* 0.118 ^a^
WT	97.40 *±* 0.200 ^a^	2.60 *±* 0.200 ^a^	0.0062 *±* 0.0004 ^b^	99.40 *±* 0.199 ^a^	57.66 *±* 0.115 ^a^
	N-NH_4_+	N-NO_3_-	pH	Conductivity	Protein
	mg/Kg	mg/Kg		mS cm^−1^	%
GNT	1597.7 *±* 1.528 ^c^	0.76 *±* 0.005 ^a^	6.32 *±* 0.000 ^b^	12.22 *±* 0.040 ^c^	0.038 *±* 0.001 ^a^
GT	1228.3 *±* 3.055 ^d^	0.60 *±* 0.004 ^b^	6.26 *±* 0.000 ^b^	11.63 *±* 0.030 ^d^	0.024 *±* 0.000 ^c^
WNT	2287.7 *±* 1.527 ^a^	0.46 *±* 0.002 ^c^	6.28 *±* 0.000 ^b^	13.78 *±* 0.142 ^a^	0.055 *±* 0.001 ^b^
WT	1865.3 *±* 5.686 ^b^	0.46 *±* 0.001 ^c^	6.74 *±* 0.000 ^a^	13.11 *±* 0.070 ^b^	0.039 *±* 0.001 ^a^
	Fibre	Carbohydrates	Lipids	Ash	Ca
	%	%	%	%	mg/Kg
GNT	0.25 *±* 0.058 ^a^	0.42 *±* 0.061 ^b^	0.24 *±* 0.031 ^b^	0.03 *±* 0.006 ^a^	425.6 *±* 2.551 ^a^
GT	0.13 *±* 0.058 ^b^	0.51 *±* 0.051 ^b^	0.21 *±* 0.028 ^b^	0.02 *±* 0.006 ^a^	446.6 *±* 2.847 ^a^
WNT	<0.01	4.38 *±* 0.020 ^a^	0.34 *±* 0.021 ^a^	0.02 *±* 0.006 ^a^	42.8 *±* 1.762 ^b^
WT	<0.01	4.26 *±* 0.025 ^a^	0.32 *±* 0.035 ^a^	0.02 *±* 0.06 ^a^	44.6 *±* 1.583 ^b^
	K	Mg	Na	P	Cd
	mg/Kg	mg/Kg	mg/Kg	mg/Kg	mg/Kg
GNT	3362.3 *±* 99.65 ^a^	631.2 *±* 2.493 ^a^	261.3 *±* 2.528 ^b^	1124.9 *±* 15.70 ^a^	0.044 *±* 0.002 ^a^
GT	3284.4 *±* 106.93 ^a^	627.3 *±* 2.115 ^a^	262.9 *±* 1.265 ^b^	1096.3 *±* 13.32 ^b^	0.042 *±* 0.002 ^a^
WNT	2112.8 *±* 80.02 ^b^	115.6 *±* 2.265 ^b^	381.7 *±* 3.155 ^a^	559.9 *±* 12.31 ^c^	0.037 *±* 0.003 ^b^
WT	2085.6 *±* 75.17 ^b^	112.3 *±* 2.351 ^b^	378.4 *±* 2.451 ^a^	556.2 *±* 8.06 ^c^	0.034 *±* 0.002 ^b^
	Ni	Pb	Zn	As	*Escherichia coli*
	mg/Kg	mg/Kg	mg/Kg	mg/Kg	UFC/g
GNT	3.533 *±* 0.156 ^a^	0.155 *±* 0.003 ^d^	7.400 *±* 0.030 ^a^	0.060 *±* 0.002 ^a^	>150,000
GT	3.167 *±* 0.328 ^a^	0.186 *±* 0.002 ^c^	7.100 *±* 0.000 ^b^	0.057 *±* 0.004 ^a^	14,000 *±* 1000.00
WNT	0.800 *±* 0.101 ^b^	0.320 *±* 0.002 ^a^	5.200 *±* 0.000 ^c^	0.059 *±* 0.006 ^a^	30 *±* 4.041
WT	0.600 *±* 0.163 ^b^	0.296 *±* 0.003 ^b^	5.067 *±* 0.015 ^d^	0.056 *±* 0.005 ^a^	28 *±* 5.00

**Table 3 plants-13-00230-t003:** Chemical composition of the water extracts of larval frass types issued from different larval diets and extracted using the E method. Concentration data refer to dry matter. The results are the average of three repeats ± standard deviation. Different letters designate significantly different values for *p* < 0.05 at the Tukey’s post hoc mean separation test. GNT = frass from Gainesville diet not treated thermally; GT = thermally treated frass from Gainesville diet; WNT = frass from the whey + seeds diet not treated thermally; WT = thermally treated frass from the whey + seeds diet.

	Water Content	Dry Matter	Total N	Organic Matter	Organic Carbon
	*%*	*%*	*%*	*%*	*%*
GNT	98.57 *±* 0.153 ^a^	1.43 *±* 0.153 ^a^	0.0018 *±* 0.0002 ^b^	99.02 *±* 0.330 ^a^	57.44 *±* 0.192 ^a^
GT	98.40 *±* 0.118 ^a^	1.60 *±* 0.118 ^a^	0.0016 *±* 0.0001 ^b^	99.35 *±* 0.225 ^a^	57.63 *±* 0.131 ^a^
WNT	98.43 *±* 0.058 ^a^	1.57 *±* 0.058 ^a^	0.0029 *±* 0.0001 ^a^	99.65 *±* 0.330 ^a^	57.80 *±* 0.191 ^a^
WT	98.43 *±* 0.058 ^a^	1.57 *±* 0.058 ^a^	0.0027 *±* 0.0001 ^a^	99.64 *±* 0.274 ^a^	57.79 *±* 0.159 ^a^
	N-NH_4_+	N-NO_3_-	pH	Conductivity	Protein
	mg/Kg	mg/Kg		mS cm^−1^	%
GNT	581.0 *±* 8.78 ^b^	0.39 *±* 0.004 ^a^	6.74 *±* 0.015 ^a^	8.88 *±* 0.041 ^c^	0.011 *±* 0.004 ^b^
GT	563.7 *±* 9.06 ^c^	0.39 *±* 0.005 ^a^	6.73 *±* 0.025 ^a^	9.14 *±* 0.040 ^b^	0.010 *±* 0.003 ^b^
WNT	820.3 *±* 10.81 ^a^	0.32 *±* 0.002 ^b^	5.19 *±* 0.053 ^b^	9.76 *±* 0.039 ^a^	0.018 *±* 0.004 ^a^
WT	696.7 *±* 9.31 ^b^	0.22 *±* 0.015 ^c^	5.31 *±* 0.061 ^b^	9.22 *±* 0.042 ^b^	0.017 *±* 0.004 ^a^
	Fibre	Carbohydrates	Lipids	Ash	Ca
	%	%	%	%	mg/Kg
GNT	0.07 *±* 0.015 ^b^	0.43 *±* 0.043 ^b^	0.37 *±* 0.008 ^a^	0.01 *±* 0.001 ^a^	218.3 *±* 4.09 ^a^
GT	0.11 *±* 0.014 ^a^	0.50 *±* 0.032 ^b^	0.39 *±* 0.012 ^a^	0.01 *±* 0.001 ^a^	121.9 *±* 2.32 ^b^
WNT	0.03 *±* 0.011 ^c^	2.48 *±* 0.046 ^a^	0.13 *±* 0.015 ^b^	0.01 *±* 0.001 ^a^	44.8 *±* 1.51 ^c^
WT	0.06 *±* 0.013 ^b^	2.39 *±* 0.041 ^a^	0.10 *±* 0.057 ^b^	0.01 *±* 0.001 ^a^	12.3 *±* 0.61 ^d^
	K	Mg	Na	P	Cd
	mg/Kg	mg/Kg	mg/Kg	mg/Kg	mg/Kg
GNT	2460.5 *±* 108.8 ^a^	216.8 *±* 0.400 ^a^	144.1 *±* 1.70 ^c^	670.7 *±* 3.51 ^a^	0.034 *±* 0.002 ^c^
GT	2243.2 *±* 110.1 ^a^	138.3 *±* 1.405 ^b^	147.6 *±* 1.60 ^c^	586.1 *±* 0.808 ^b^	0.059 *±* 0.003 ^a^
WNT	1253.0 *±* 89.5 ^b^	96.4 *±* 1.200 ^c^	224.5 *±* 1.42 ^a^	361.0 *±* 2.69 ^c^	0.055 *±* 0.004 ^a^
WT	1152.8 *±* 103.8 ^b^	81.4 *±* 0.529 ^d^	207.6 *±* 2.43 ^b^	314.5 *±* 5.00 ^d^	0.041 *±* 0.002 ^b^
	Ni	Pb	Zn	As	
	mg/Kg	mg/Kg	mg/Kg	mg/Kg	
GNT	1.723 *±* 0.031 ^a^	0.164 *±* 0.04 ^a^	3.585 *±* 0.004 ^c^	0.217 *±* 0.018 ^b^	
GT	0.695 *±* 0.007 ^b^	0.224 *±* 0.03 ^a^	5.901 *±* 0.013 ^a^	0.253 *±* 0.019 ^b^	
WNT	0.235 *±* 0.003 ^c^	0.081 *±* 0.03 ^b^	4.023 *±* 0.007 ^b^	0.214 *±* 0.022 ^b^	
WT	0.168 *±* 0.006 ^d^	0.051 *±* 0.04 ^b^	2.420 *±* 0.016 ^d^	0.335 *±* 0.022 ^a^	

## Data Availability

Data are available upon request.

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
