# Peer review of "Germination Behavior and Geographical Information System-Based Phenotyping of Root Hairs to Evaluate the Effects of Different Sources of Black Soldier Fly (Hermetia illucens) Larval Frass on Herbaceous Crops"

_plants, 2024, doi:10.3390/plants13020230_

Round 1
Reviewer 1 Report
Comments and Suggestions for Authors
Different diets and thermal treatments given to black soldier fly larvae affected their frass' chemical compositions and plant growth differently. The Gainesville diet at 25% concentration led to better plant growth. Frass from the whey-seed diet suppressed Escherichia coli and had a phytotoxic effect on germinating plants. A new GIS-based method was proposed to determine root hair surface area for efficient root hair phenotyping in frass evaluation. I have a few suggestions for the manuscript. The research mainly focused on frass's chemical and microbiological parameters, but it did not extensively investigate the mechanisms underlying the observed effects on plant growth and root functionality. However, overall, the work is good with a beginning, middle, and end!
Firstly, I think that the GIS abbreviation could be spelled out as “Geographic Information System”, but this can be discussed with the editorial team. Additionally, some figures (such as figures 3, 7, and 9) could benefit from being in color. Another issue that was observed is that axis and their accompanying texts and numbers could be made more evident.
It is recommended to italicize the scientific names such as Triticum durum Desf. (cv. Ciclope), barley (Hordeum vulgare L.) (cv. Laureate), and cress (Lepidium sativum L.).
Comments on the Quality of English Language
By and large, the manuscript requires a proofreading. Some words were confused and misspelled (such as "than" instead of "then", "numer" instead of "number", "treaements" instead of "treatments", and "Results" instead of "Results", among others), and some sentences were unclear.
Author Response
We thank the reviewer for his comments and suggestions. We modified the manuscript as required and included some changes suggested by another reviewer as well.
Best regards,
Mariana Amato

Reviewer 2 Report
Comments and Suggestions for Authors
The article, titled "Germination Behavior and GIS-Based Phenotyping of Root Hairs in Evaluating Black Soldier Fly Larval Frass Effects on Herbaceous Crops," explores the impact of diverse frass sources on plant-soil relations. Investigating insect farming by-products as potential fertilizers, the research assesses phytotoxicity from various frass sources obtained through black soldier fly breeding. Results reveal varying chemical and microbiological parameters based on diet and thermal treatment. Effects on cress and wheat include concentration-dependent phytotoxicity and diet-specific influences on germination and root development. Barley seedling trials demonstrate impacts on emergence, chlorophyll content, and root mass. The paper proposes a GIS-based procedure for rapid root hair phenotyping.
While the study holds particular significance and has the potential to contribute significantly to the field, it is recommended to thoroughly address and consider various general aspects before proceeding with the manuscript's publication:
1. In the title, before using any abbreviations, they should be illustrated. For example, 'GIS' (Geographic Information System). Additionally, the full stop should be removed from the title.
2. The abstract currently has nearly 408 words, which exceeds the recommended limit of 200 words according to MDPI guidelines. It needs to be reduced, and improvement could be achieved by addressing the following points:
a. Present a concise overview of the study's methodology, clarify the main goal of product development, and conclude by examining both favorable and unfavorable consequences.
b. Attention is required to rectify writing issues, including the addition of spaces, such as in unit separations.
c. Emphasize quantitative results to offer more precise and concrete evidence that supports the conclusions.
d. Ensure that the keywords align with the Plants guidelines.
3. Introduction:
a. The authors should discuss existing limitations in literature and highlight the novelty of their work in addressing these limitations.
b. Additionally, the authors should emphasize the potential applications of the prepared GIS-based procedure. They should highlight the main advantages of this process.
c. English issues throughout the entire manuscript should be thoroughly revised by experts.
d. Attention should be given to the numbering of references using any available tools. For instance, Ref 13 appears before Ref 12. Please double-check and correct this throughout the entire manuscript.
4. Results.
a. Please revise the table format to align with the Plants template. Additionally, it is understood that the letters a, b, c, d in the table are meant to indicate the significance of measurements and should be in superscript form.
b. Standard deviation, as well as table notes, should be provided for all tables.
c. Please double-check for typographical and grammatical issues, including superscripts, subscripts, etc.
d. The authors should consider using the same font and color for all figures and tables to homogenize and facilitate comparison.
Figure axes should be thickened, and numbers should be in bold to enhance their visibility and readability.
e. Line 161: Figure 1 shows cress germination indices GERIS (shoot-based germination index) and GERIS (root-based germination index).?
5. Experimental
a. The "Materials and Methods" section lacks credibility and should be appropriately referenced to confirm its reliability. The absence of references in this section raises concerns about the study's validity, bordering on being perceived as an invention.
b. The characterization techniques should address the methodology of preparing samples for measurements and describe the equipment used in the study.
c. The chemical composition should be characterized using advanced techniques such as XRD, FTIR, and UV-Vis for a more comprehensive analysis.
6. Conclusion
a. This being the first study of its kind without a conclusion, I recommend that the authors allocate sufficient time and attention to revise, restructure, and redesign the manuscript. They should provide a thorough conclusion section that highlights the main results, addresses limitations, and outlines future perspectives for further studies, among other relevant aspects.
Comments on the Quality of English Language
Extensive editing of English language required
Author Response
We thank the reviewer for his comments and suggestions. We modified the manuscript as required and included some changes suggested by another reviewer as well.
Best regards
Mariana Amato

Round 2
Reviewer 2 Report
Comments and Suggestions for Authors
The revised version provided by the authors was so challenging to revise. Using Track Changes makes reading and revisiting uncomfortable. Additionally, the attached document containing the responses had very small font sizes, and the document is disabled for editing. However, after careful revision and time investment, the authors are appreciated for their efforts and the time dedicated to improving their manuscript. There are still some points that need minor revision, as follows:
a. The abstract is still lengthy; however, according to the authors' response, the 300 words align with the editorial standards. If this is the case, it's great news for the authors.
b. English issues throughout the entire manuscript should be thoroughly revised by experts.
c. There are still some issues with the reference format. For example, on line 73, the reference is cited as '(for a review see [2)],' and this needs correction.
d. The conclusion section should be at the end of the manuscript.
Comments on the Quality of English Language
English issues throughout the entire manuscript should be thoroughly revised by experts.
Author Response
The revised version provided by the authors was so challenging to revise. Using Track Changes makes reading and revisiting uncomfortable. Additionally, the attached document containing the responses had very small font sizes, and the document is disabled for editing. However, after careful revision and time investment, the authors are appreciated for their efforts and the time dedicated to improving their manuscript.
We apologize. We were not aware the reviewer would not have access to the file with accepted changes and the word format of the response file. We are grateful to the reviewer for her/his work, and are using larger fronts for this response.
There are still some points that need minor revision, as follows:
- The abstract is still lengthy; however, according to the authors' response, the 300 words align with the editorial standards. If this is the case, it's great news for the authors.
While the “Instructions for authors” indicate a limit of 200 words for the abstract, we found that abstracts of published papers are longer than that in many instances, and may exceed 300 words (e.g. 345 words for the abstract of the most recent paper on the Plants’ website: “Plants. 2024; 13(2):172. https://doi.org/10.3390/plants13020172”).
- English issues throughout the entire manuscript should be thoroughly revised by experts.
We had the manuscript checked for English issues and many sentences were rephrased throughout the manuscript
- There are still some issues with the reference format. For example, on line 73, the reference is cited as '(for a review see [2)],' and this needs correction.
We changed the sentence to:
“First results on plant growth are encouraging as reported by Guidini Lopes et al. [2] in a recent review. However, ….”
- The conclusion section should be at the end of the manuscript.
The conclusion section was moved to the end of the manuscript and the “Materials and methods” section was re-numbered accordingly.